# Increase of Circulating Monocyte–Platelet Conjugates in Rheumatoid Arthritis Responders to IL-6 Blockage

**DOI:** 10.3390/ijms23105748

**Published:** 2022-05-20

**Authors:** Anaís Mariscal, Carlos Zamora, César Díaz-Torné, Mᵃ Àngels Ortiz, Juan José de Agustín, Delia Reina, Paula Estrada, Patricia Moya, Héctor Corominas, Sílvia Vidal

**Affiliations:** 1Immunology Department, Hospital de la Santa Creu i Sant Pau, Biomedical Research Institute Sant Pau (IIB Sant Pau), 08041 Barcelona, Spain; amariscal@santpau.cat; 2Laboratory of Inflammatory Diseases, Hospital de la Santa Creu i Sant Pau, Biomedical Research Institute Sant Pau (IIB Sant Pau), 08041 Barcelona, Spain; carlosza@g86mail.com (C.Z.); mortiz@santpau.cat (M.À.O.); 3Rheumatology Department, Hospital de la Santa Creu i Sant Pau, 08041 Barcelona, Spain; cdiazt@santpau.cat (C.D.-T.); pmoyaa@santpau.cat (P.M.); hcorominas@santpau.cat (H.C.); 4Rheumatology Department, Hospital Vall d’Hebrón, 08035 Barcelona, Spain; jjagor@hotmail.com; 5Rheumatology Department, Hospital Moisès Broggi, Sant Joan Despí, 08970 Barcelona, Spain; deliareinasanz@gmail.com (D.R.); paulavestradaa@gmail.com (P.E.)

**Keywords:** platelets, monocytes, rheumatoid arthritis, tocilizumab, immune modulation, inflammation, immunity

## Abstract

Platelets (PLT) bind to a significant percentage of circulating monocytes and this immunomodulatory interaction is increased in several inflammatory and autoimmune conditions. The therapeutic blockage of IL-6 with Tocilizumab (TCZ) alters PLT and the phenotype and function of monocytes in rheumatoid arthritis (RA). However, the relationship between monocyte–PLT conjugates (CD14+PLT+) and clinical and immunological variables and the regulation of this interaction by IL-6 blockage are still unknown. Here, we compared the presence of monocyte–PLT conjugates (CD14+PLT+) and membrane CD162 expression using flow cytometry, and, by ELISA, the markers of PLT activation (sCD62P and sCD40L) in healthy donors (HD) and patients with long-standing RA before TCZ (baseline). We found higher percentages and absolute counts of CD14+PLT+, and higher plasmatic levels of sCD62P and sCD40L but lower CD162 expression on monocytes from RA patients than those from HD. Additionally, the levels of CD14+PLT+ inversely correlated with inflammatory parameters. Interestingly, 95% of patients with lower percentages of CD14+PLT+ and only 63% of patients with higher percentages of CD14+PLT+ achieved a EULAR-defined response at four weeks (*p* = 0.036). After TCZ, the percentage of CD14+PLT+ increased in 92% of RA patients who achieved 12 w-remission (*p* < 0.001). Our results suggest that the binding of PLTs has a modulatory effect, accentuated by the increased binding of PLTs to monocytes in response to the therapeutic blockage of IL-6.

## 1. Introduction

Rheumatoid arthritis (RA) is an inflammatory joint disease characterized by persistent systemic inflammation, with extra-articular manifestations that can affect up to 41% of patients [1]. Neutrophils, mast cells, T and B lymphocytes, and monocytes/macrophages are involved in the development of inflammation in RA [2]. In particular, monocytes play an important role in the initiation, maintenance and the degree of activity of synovial inflammation [3]. In RA, activated monocytes possess increased adherent abilities [4]. They massively infiltrate inflammatory sites, where they differentiate into tissue macrophages and type A synoviocytes [5,6]. The activation of these cells can also lead to the production of TNF-α, IL-1, IL-6, and IL-8, which are effector molecules involved in inflammation [4,7].

Platelets (PLTs) are also involved in inflammatory processes through the secretion of inflammatory mediators or through a direct interaction with leukocytes [8,9,10,11]. PLTs circulate in an inactive state and can respond to a wide variety of stimuli, releasing cytokines, chemokines, and growth factors that are stored in granules [12,13,14,15]. PLTs from RA patients express higher P-selectin (CD62P) levels and produce greater amounts of soluble CD40 ligand (sCD40L), both of which are markers of PLT activation [16,17]. One of the factors involved in this increased PLT activation is anti-citrullinated protein antibodies (ACPA) [13].

Circulating activated PLTs can bind to other PLTs and leukocytes, setting off a cascade of events that contribute to the development, evolution, and resolution of systemic inflammatory response. PLT adhesion to circulating leukocytes enhances leukocyte rolling on the endothelial wall, facilitating the transport of PLTs to joint space [18,19]. In healthy donors (HD), P-selectin glycoprotein ligand 1 (PSGL-1 or CD162) on leukocytes seems to be the main molecule involved in the interaction of PLTs and leukocytes when it binds to CD62P on activated PLTs [20,21]. However, other ligands can be involved in the binding of PLTs to monocytes: GPIb-CD11b [22], CD40-CD40L [23], GPIIb/IIIa-CD11/CD18 [24], EMMPRIN (CD147/basigin)-CD147/GPIV [25,26], TREM-1-TREM-1 ligand [27] and PADGEM [28]. We previously described an increased percentage of monocytes with bound PLTs in SLE patients. However, SLE monocytes have a decreased expression of CD162, suggesting that other molecules contribute to this binding [29]. Other authors have also reported the down-regulation of CD162 on myeloid innate cells in an inflammatory context or after leukocyte activation [30,31,32,33]. In HD, monocyte–PLT interaction decreases monocyte apoptosis, increases the expression of co-stimulatory molecules and Fc gamma receptors, and leads to a phenotypic shift in CD14+CD16- towards CD14+CD16+ and a monocyte differentiation into macrophages [29,34,35,36]. Furthermore, monocytes with bound PLTs have increased phagocytosis ability [29]. PLTs can also dampen inflammatory responses by increasing IL-10 and decreasing IL-6 and TNF-α release by monocytes [37,38]. Like other authors, we have reported an increased percentage of monocytes with bound PLTs in several inflammatory and autoimmune conditions, including RA [19,26,29,32,39,40,41,42,43]. Depending on the disease, these monocyte–PLTs aggregates can play pro- [44], or anti-inflammatory roles [29,32,45]. Specifically, PLTs from RA patients can induce HD monocytes to synthesize pro-inflammatory cytokines (TNFα and IL-6) through CD147 engagement [26].

Although RA methotrexate (MTX), a conventional synthetic disease-modifying anti-rheumatic drug (DMARDs), is the first-line treatment, not all patients achieve remission. Tocilizumab (TCZ) is indicated in patients who have failed two courses with conventional synthetic DMARDs or who have adverse prognostic markers [46]. TCZ is a humanized anti-human IL-6 receptor antibody that blocks the binding of IL-6 to its receptor, thereby preventing signalling. It has been shown that TCZ induces apoptosis, downregulates the expression of CD80 and suppresses the expression of IL-6 on stimulated monocytes from HD [47]. Additionally, in more than 10% of patients, PLTs have been affected by TCZ [48,49]. Therefore, the main goal of this study is to decipher whether a new mechanism of action of TCZ on RA regulates the binding of PLT to monocytes. First, we analysed monocyte–PLT conjugates, as well as markers of PLT activation, in a cohort of long-lasting RA patients that are refractory to standard treatment with DMARDs. Second, we studied the association between the frequency of monocytes with bound PLTs and the clinical and laboratory features of RA patients before TCZ treatment. Third, we investigated whether monocytes with bound PLTs can function as biomarkers of TCZ treatment response. Finally, we clarified the functional consequences of both IL-6 signalling and TCZ treatment over PLTs binding to monocytes in in vitro cultures.

## 2. Results

### 2.1. Circulating CD14+PLT+ in RA Patients

We observed higher percentages and absolute counts of circulating CD14+PLT+ in RA than in HD (Figure 1A,B and Appendix A). However, we did not find differences in absolute counts of monocytes between RA patients and HD (201.08 ± 77.12 CD14+/µL in RA patients vs. 208.44 ± 54.23 CD14+/µL in HD; *p* = 0.8). We also found a direct correlation between the percentage and absolute counts of CD14+PLT+ in both RA patients and HD (Figure 1C). When we compared membrane CD162 (mCD162) expression, we found that monocytes from RA patients had a lower expression of mCD162 than HD (Figure 1D and Appendix A). In addition, mCD162 expression directly correlated with the percentage of CD14+PLT+ in HD (Figure 1E) but correlated inversely in RA patients (Figure 1F). mCD162 expression did not correlate with absolute counts of circulating CD14+PLT+ in RA patients (r = −0.199; *p* = 0.284) or in HD (r = 0.34; *p* = 0.306). Monocytes with or without bound PLTs had similar mCD162 expressions (Figure 1G). There were no differences in soluble CD162 (sCD162) between RA patients and HD (Figure 1H).

We did not find any association between the percentage or absolute counts of CD14+PLT+ and the plasmatic concentrations of IL-6, sIL-6R, IL-10, IL-22, LBP and VEGF (data not shown). However, we found that the percentage, but not the absolute counts, of CD14+PLT+ correlated with plasmatic IL-17 levels in RA patients but not in HD (r = 0.505; *p* = 0.006 and r = 0.163; *p* = 0.407 respectively).

We next analysed surrogate soluble markers of PLT activation: sCD40L and sCD62P [16]. Although we found that both molecules were higher in RA patients than in HD (Appendix A), no association was found between sCD40L or sCD62P and the percentage of CD14+PLT+ in RA patients or in HD (Appendix A). In addition, we did not observe any association between sCD40L and sCD62P in RA patients or in HD (Appendix A).

### 2.2. Relationship between CD14+PLT+ and mCD162 Expression on Monocytes with Clinical RA Parameters

We found no association between the percentages or the absolute counts of CD14+PLT+ and mCD162 expression on monocytes and DAS28, SDAI, or CDAI scores before TCZ treatment. Since those disease activity indexes integrate different parameters, we then analysed each parameter independently. We found that absolute counts of CD14+PLT+ were inversely correlated with inflammatory markers CRP (Figure 2A) and ESR (Figure 2B), but CD14+PLT+ percentages did not correlate with CRP and ESR (r = −0.4; *p* = 0.054 with CRP and r = −0.33; *p* = 0.09 with ESR). mCD162 expression on monocytes correlated with joint US scores (Figure 2C) and health assessment questionnaire scores (HAQ) (Figure 2D).

We also observed a correlation between sCD40L and RF titers (r = 0.544; *p* = 0.013). In addition, we found that sCD40L levels between ACPA+ and ACPA- patients (839 (389.1–1794) pg/mL in ACPA+ vs. 332.9 (64.7–1551) pg/mL in ACPA-; *p* = 0.08) were not significantly different.

### 2.3. Relationship between CD14+PLT+ and mCD162 Expression on Monocytes with Clinical Outcome in RA Patients after TCZ Treatment

We found that the percentage of CD14+PLT+ increased, while mCD162 expression on monocytes decreased 4 and 12 weeks after initiating TCZ treatment (Figure 3A,B). sCD62P also decreased at four weeks (Appendix A), while sCD40L did not change after TCZ treatment (data not shown).

The segregation of patients into those with low or high percentages or absolute counts of CD14+PLT+ was established by calculating cut-offs with a confidence interval of 95% for HD values, as explained in the Material and Methods section. A higher percentage of patients with lower percentages or absolute counts of CD14+PLT+ at baseline achieved a good (2) or moderate (1) EULAR response compared to patients with higher percentages or absolute counts of CD14+PLT+ (%CD14+PLT+ low: HR = 2.68 when compared to %CD14+PLT+ high) (Figure 3C and data not shown). A decreased percentage of CD14+PLT+ at four weeks after TCZ was found in patients who did not achieve remission at 12 weeks. In contrast, an increased percentage of CD14+PLT+ at four weeks after TCZ was found in patients who achieved remission at 12 weeks (Figure 3D). A decreasing mCD162 expression four weeks after TCZ was found only in patients who did not achieve remission (Figure 3E). Regarding sCD62P, there were no significant changes in patients who did not achieve remission, while decreasing changes were found in patients who achieved remission (Appendix A). Interestingly, in the group of patients who achieved remission, changes in CD14+PLT+ were inversely correlated with changes in sCD62P (r = −0.582, *p* = 0.04). No correlation was found in the non-responder group (data not shown).

We also analysed the association between changes in the percentage of CD14+PLT+, mCD162 expression on monocytes and sCD62P at four weeks with changes in clinical and laboratory parameters at four weeks. We found that changes in CD14+PLT+ correlated inversely with changes in DAS28, ESR, and with swollen joint count (SJC) (Figure 4A,C). We did not find any association between changes in mCD162 expression on monocytes or plasmatic sCD62P levels and changes in clinical and laboratory parameters (data not shown).

### 2.4. IL-6 and TCZ Effects over PLT Binding to Monocytes, mCD162 Expression on Monocytes and PLT Activation

To analyse the mechanism of TCZ on PLT binding to monocytes, we stimulated whole blood from HD with LPS, rhIL-6, or/and TCZ. We found an increased percentage of CD14+PLT+ when we stimulated whole blood with LPS or with rhIL-6, without synergic effects when both stimuli were present (data not shown). When we added TCZ after LPS or rhIL-6 stimulus, the percentage of CD14+PLT+ was not different from LPS or rhIL-6 conditions (data not shown). However, when we added TCZ before LPS or rhIL-6 stimulus, the percentage of CD14+PLT+ was significantly lower than LPS or rhIL-6 conditions (Figure 5A, Appendix A).

We also found that mCD162 expression on monocytes decreased in the presence of LPS (Figure 5B, Appendix A). No differences in mCD162 expression on monocytes were found between LPS alone and LPS when we had added TCZ beforehand (Figure 5B, Appendix A) or after LPS stimulus (data not shown). The addition of TCZ before rhIL-6 decreased mCD162 expression on monocytes (Figure 5B, Appendix A). We did not find differences in the percentage of activated free PLTs (CD41a+CD62P+) between the different stimulations (data not shown).

## 3. Discussion

Our present findings suggest that, as a result of exposure to an inflammatory environment, more monocytes bound to PLTs but had decreased mCD162 expression. After the administration of TCZ, the percentage of CD14+PLT+ increased in RA patients who achieved remission after 12 weeks post-TCZ treatment. Interestingly, patients with a lower percentage of CD14+PLT+ before TCZ treatment achieved EULAR-defined response to a greater extent.

Our work showed that RA patients had increased percentages and absolute counts of CD14+PLT+ cells and increased levels of soluble markers of PLT activation. We also found an inverse association between CD14+PLT+ and the inflammatory parameters CRP and ESR. These findings suggest that an inflammatory environment contributes to the binding of PLT to monocytes with a regulatory effect. Whereas, like other authors, we have not found any statistical association between PLT activation markers and disease activity [50], Lee et al. found a correlation between sCD62P levels and disease activity, CRP and ESR [51]. This discrepancy between studies may be due to differences in cohort characteristics. In fact, our cohort had a longer disease duration and higher disease activity but a lower percentage of RF- and ACPA-positive patients, as well as a lower percentage of patients in MTX treatment.

We found a significantly lower expression of mCD162 on monocytes in RA patients, despite them having more CD14+PLT+ cells. This finding suggests that, in addition to mCD162, other molecules on the surface of monocytes participate in PLT–monocyte interaction during inflammation. This fact is not exclusive to RA, and we previously found an increased percentage of CD14+PLT+ and a lower expression of mCD162 on monocytes in other chronic inflammatory diseases, such as ulcerative colitis and SLE [29,32]. Some authors have described a mCD162 downregulation on monocytes in the presence of endotoxin in HIV patients or after leukocyte activation [4,30,31,33,52]. There are three possible explanations for these findings. First, the lower expression of mCD162 may have been due to a misdetection by the binding of PLTs that hide this molecule. This is unlikely because RA monocytes with or without bound PLTs had a similar mCD162 expression in our experiments. Second, the lower expression of mCD162 may be related to the activation state of monocytes [4,52] and the shedding of this molecule from their cell surface. Particularly, IL-8 and GM-CSF induce ADAM10 to shed CD162 on the cell-surface [33,53]. However, despite the lower mCD162 expression on RA monocytes, we did not find differences in sCD162 levels, the product of mCD162 shedding, between RA and HD. Since we measured sCD162 on plasma samples and supernatants from whole-blood cultures, we cannot rule out sCD162 consumption and/or degradation in our experimental system. Third, the lower expression of mCD162 on RA monocytes may have been due to reduced transcription. Therefore, we cannot discard the possibility that mCD162 downregulation is secondary to activation and shedding or to reduced transcription in RA monocytes.

The follow-up analysis of patients during our study showed that TCZ treatment exacerbated the differences between RA patients and HD, increasing the percentage of CD14+PLT+ and decreasing plasmatic sCD62P and the mCD162 expression on monocytes. Again, these results support the idea that the binding of PLT to monocytes is a regulatory mechanism. In line with this, we found that, in patients in remission at 12 weeks, the percentage of CD14+PLT+ increased and correlated with the decrease in sCD62P. This result seems to indicate that the binding of non-activated PLT to monocytes has beneficial consequences. However, it is possible that other DMARDs have different effects on monocyte and PLT interaction due to characteristic target cells in the therapy mechanisms of action. In line with this, Veale et al. reported a decrease in sCD62P after sulphasalazine. Although Bunescu et al. reported a decrease in monocyte activation after MTX but not after TNF blocker treatment, they did not find changes in monocyte–PLT aggregates after either of the treatments [19,50].

Some studies have revealed different consequences after PLT–monocyte interaction. Thus, the specific binding of PLTs from RA patients to HD monocytes through CD147 increases TNFα and IL-6 expression on monocytes [26]. In contrast, we have reported that the co-cultures of PLT from HD with synovial fluid cells from RA patients reduced inflammatory cytokine production and increased IL-10 production [54]. Thus, it is likely that PLT can exert different effects, depending on the environment in which they have been “educated” and the cellular target for their binding. We speculate that the binding of PLT to monocytes after TCZ, with less inflammation and activation of PLT and monocytes, may have different consequences in comparison to binding before TCZ treatment.

When we stimulated in vitro whole blood from HD with LPS or rhIL-6, the percentage of CD14+PLT+ increased. These results would explain ex vivo observations in patients before TCZ therapy. It is possible that IL-6 modifies the activation status of monocytes. Another possibility is that IL-6 induces CD16 expression on monocytes and PLTs would therefore preferentially bind to CD16+ monocytes [34,37,51]. This is unlikely in our settings because we did not find differences in CD16 expression on monocytes cultured for 28 h with or without rhIL-6 (data not shown). A detailed kinetics with different times of culture will clarify this point. IL-6 might also activate PLTs in RA patients. However, we observed a higher binding of PLT to monocytes without higher PLT activation, suggesting that both monocyte activation and phenotype changes are crucial in the interaction with PLT.

Further conclusions regarding the involvement of IL-6 in PLT–monocyte interaction were obtained when we blocked in vitro IL-6 signaling with TCZ. The addition of TCZ to the culture prior to, but not after, LPS or rhIL-6 prevented the increase in CD14+PLT+, suggesting that TCZ has a preventive effect. Nevertheless, we are aware of the difficulty of extrapolating the results of in vitro 28 h experiments to in vivo RA patients after 12 weeks of follow-up [4,16,55]. Additionally, differences in both monocytes and PLTs between RA patients and HD could also explain this difference, as monocytes and PLTs from RA patients are activated [4,16,55]. Further experiments with RA samples will reveal if the in vitro effects of TCZ over monocyte–PLT conjugates are equivalent in RA and HD. Interestingly, in RA patients, despite the high levels of CD14+PLT+ in half of the patients before TCZ, these increased further after initiating TCZ treatment. A possible explanation for the high levels at baseline in this group is that CD14+PLT+ is an induced regulatory mechanism to try to control inflammation.

In line with our findings with TCZ treatment, mCD162 expression decreased when we added TCZ prior to rhIL-6 culture. However, we did not reproduce this effect when TCZ was added before or after LPS culture. These results support the idea that other molecules, in addition to IL-6, regulate mCD162 in an inflammatory context. Hashizume et al. showed that the stimulation of whole blood with IL-6 for two hours downregulated mCD162 expression [53]. We did not reproduce the results of Hashizume et al. when stimulating with rhIL-6. Differences in culture timing could explain this discrepancy, as our experimental system has a 28-h duration.

To avoid the interference of multiple mechanisms involved in different phases of RA, we have only included patients with a long disease duration. This restriction may have biased the cohort to a group with more systemic inflammation and refractoriness to previous DMARDs. This restriction has also limited the number of patients. To generalize our findings, a larger cohort, including early-onset patients and patients without previous treatments, should be included. Furthermore, our observations should be completed with additional experiments that could explain the mechanism by which TCZ treatment increases CD14+PLT+, as well as the functional consequences of this rise. Nevertheless, in addition to the description of a new mechanism in the pathology of RA, the role of monocyte–PLT conjugates in the modulation of inflammation, our results suggest that the percentage of CD14+PLT+ can be applied as a biomarker of early remission to TCZ treatment.

## 4. Materials and Methods

### 4.1. Patients

Peripheral blood samples from 35 RA patients and 15 HD were collected in BD vacutainer CPT tubes containing heparin (BD, Franklin Lakes, NJ). RA diagnosis was based on the American College of Rheumatology diagnostic criteria for RA [56]. All RA patients were refractory to standard treatment with DMARDs, including MTX. TCZ treatment was started following European and Spanish guidelines [57,58]. The demographic, clinical and laboratory data of RA patients enrolled in this study are shown in Table 1. This study was approved by the ethics committee of Hospital de la Santa Creu i Sant Pau. In accordance with the Declaration of Helsinki, written consent was obtained from all patients.

Patients were treated with 8 mg/kg of TCZ every four weeks. Response to TCZ therapy was evaluated at 12 weeks. Blood samples and clinical data were collected one hour prior to the first infusion (baseline) and one hour prior to the respective infusions at weeks four (4 w) and 12 (12 w) after initiating treatment. Laboratory analyses for all visits included a hemogram, erythrocyte sedimentation rate (ESR), C-reactive protein (CRP), rheumatoid factor (RF), and anti-cyclic citrullinated peptide antibodies (ACPAs). Joint ultrasound (US) scores for all visits were evaluated using power doppler and grey scale [59]. Clinical data were collected at each visit. The disease activity score (DAS) 28 was calculated with ESR values. Simplified Disease Activity Index (SDAI), and Clinical Disease Activity Index (CDAI) were also calculated [60]. EULAR response criteria were used to classify patients as good, moderate or non-responders after 12 weeks of TCZ [61]. A good response patient (EULAR 2) had a ΔDAS28 > −1.2 and DAS28 final ≤ 3.2. A non-response patient (EULAR 0) had a ΔDAS28 ≤ −0.6 or −1.2 ≤ ΔDAS28 ≤ −0.6 and DAS28 final > 5.1. The rest of the patients (EULAR 1) had a moderate response.

### 4.2. Peripheral Blood Staining and Flow Cytometry Analysis

Peripheral blood cells (100 µL) were incubated for fifteen minutes at room temperature in the dark with anti-CD14-APC (clone 18D11), -CD41a-FITC (clone HIP8) (ImmunoTools, Friesoythe, Germany), and -CD162-PE (clone KPL-1) (Biolegend, San Diego, CA, USA) mAb. Red blood cells were lysed and white cells were fixed using BD FACS lysing solution (BD Biosciences, San Jose, CA, USA) to be analysed by flow cytometry.

Monocyte–platelet conjugates were identified as CD14+CD41a+ cells (CD14+PLT+). Surface expression of CD162 was analysed on gated CD14+ monocytes, as well as on CD14+PLT- and CD14+PLT+. Samples were acquired with the MACSQuant Analyzer 10 flow cytometer (Miltenyi Biotec GmBh, BergischGaldbach, Germany). We calculated the percentage of positive cells and mean fluorescence intensity (MFI) of each individual marker using MACSQuant Analysis Software (v.2.13.1).

### 4.3. Plasma Analysis of RF, ACPAs, and Cytokines

RF titration was determined by nephelometry (Beckman ICS II, Beckman Coulter). ACPA titre was determined by EliA test using UniCAP (Phadia Laboratory Systems, Uppasala, Sweden). Plasma was tested for IL-17, IL-22, VEGF, sCD40L (Peprotech, London, UK), soluble P-Selectin (sCD62P), soluble CD162 (sCD162) (R&D Systems, Minneapolis, MN), Lipopolysaccharide binding protein (LBP) (Hycult Biotech, Uden, The Netherlands), IL-10, IL-6 and soluble IL-6 receptor (sIL-6R) (ImmunoTools), using specific ELISAs according to the manufacturer’s instructions. All cytokines were quantified with standard curves provided by manufacturer. The detection limits were: 31.2 pg/mL for IL-17, 15.6 pg/mL for IL-22, 31.2 pg/mL for VEGF, 15.6 pg/mL for sCD40L, 125 pg/mL for sCD62P, 31.3 pg/mL for sCD162, 4.4 ng/mL for LBP, 16 pg/mL for IL-10, 3 pg/mL for IL-6 and 230 pg/mL for sIL-6R.

### 4.4. Peripheral Blood Cultures

Heparinized blood was diluted with RPMI 1640 medium (supplemented with 2 mM glutamine, 100 U/mL penicillin, and 100 mg/mL streptomycin (BioWhittaker, Verviers, Belgium)) at a proportion of 1:1.4. Blood cells were stimulated with ultrapure TLR4 agonist lipopolysaccharide (LPS) (Invivogen, San Diego, CA, USA) at 10 pg/mL for four hours before adding 50 ng/mL of recombinant human IL-6 (rhIL-6) (ImmunoTools) for 12 h. TCZ was added at 100 µg/mL 30 min prior to LPS (conditions pre) or after rhIL-6 (conditions post). Culture conditions were then as follows: TCZ (pre) + LPS + rhIL-6 and LPS + rhIL-6 + TCZ (post). As control conditions, we included: (a) absence of stimulation, (b) LPS alone, (c) rhIL-6 alone, (d) TCZ alone, (e) rhIL-6+ TCZ (post), (f) TCZ (pre) + LPS, (g) TCZ (pre) + rhIL-6, (h) LPS + TCZ (post), (i) LPS + rhIL-6. After culture, supernatants were collected and kept at −20 °C until used. A total of 100 µL of cultured whole blood from each condition was stained with anti-CD14-PECy7 clone M5E2 (BD Biosciences), anti-CD16 AF647 clone 3G8, anti-CD162 (BioLegend), anti-CD41a-FITC, anti-CD62P-APC clone HI62P (Immunotools). Red blood cells were lysed and white cells were fixed using BD FACS lysing solution (BD Biosciences) to be analysed by flow cytometry. Activated PLTs were identified as CD41a+CD62P+. Samples were acquired with the MACS-Quant Analyzer 10 flow cytometer (Miltenyi Biotec). We calculated the percentage of positive cells and the expression (MFI) of each individual marker using MACSQuant Analysis Software.

### 4.5. Statistical Analysis

Statistical analyses were performed using Graph Pad Prism 7 software (v.7.00). The Kolmogorov–Smirnov test was applied to test the normal distribution of the data. Variables with a normal distribution were reported as mean ± standard deviation (SD) and the variables with a non-normal distribution were reported as median (interquartile range) (IQR). The *t*-test and paired *t*-test were used for the comparison of independent and related variables with normal distribution, respectively. The Mann–Whitney test and Wilcoxon test were used for the comparison of independent and related variables with non-normal distribution, respectively. The long-rank Mantel–Cox test was used to analyse differences in EULAR response during the follow-up period. *p* values lower than 0.05 were considered significant.

## Figures and Tables

**Figure 1 ijms-23-05748-f001:**
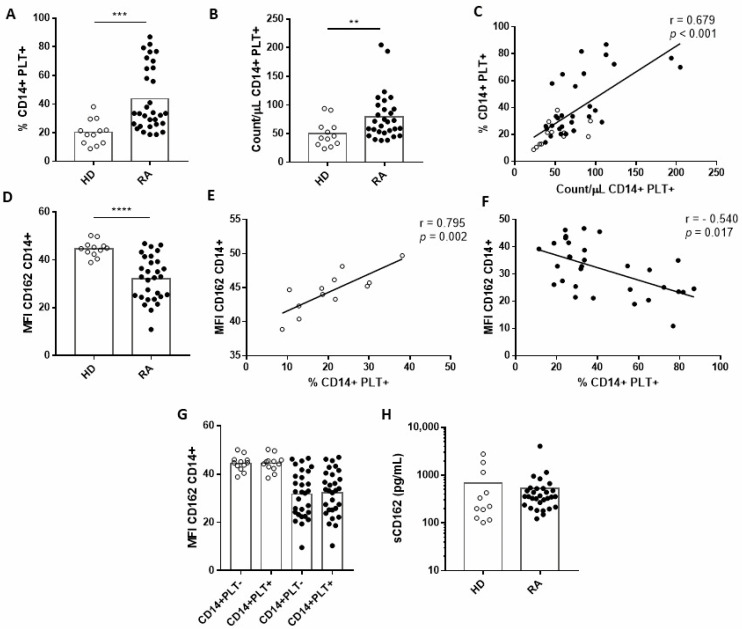
Monocytes with bound platelets and CD162 expression on monocytes from healthy donors (HD) and rheumatoid arthritis (RA) patients. (**A**) Percentage and (**B**) absolute numbers of monocytes with bound platelets (CD14+PLT+) in HD and RA patients. (**C**) Correlation between percentage and absolute number of CD14+PLT+ in HD and RA patients. (**D**) CD162 expression on monocytes from HD and RA patients. (**E**) Correlation between CD162 expression and percentage of CD14+PLT+ in HD and (**F**) RA patients. (**G**) CD162 expression on CD14+PLT- or CD14+PLT+ in HD and RA patients. (**H**) Quantification of soluble CD162 (sCD162) in plasma from HD and RA patients. White circles represent HD, and black circles represent RA patients. Statistical analysis was performed using the Mann–Whitney test for comparisons between HD and RA and Spearman’s correlation for correlation analysis. ** *p* < 0.01, *** *p* < 0.001, **** *p* < 0.0001.

**Figure 2 ijms-23-05748-f002:**
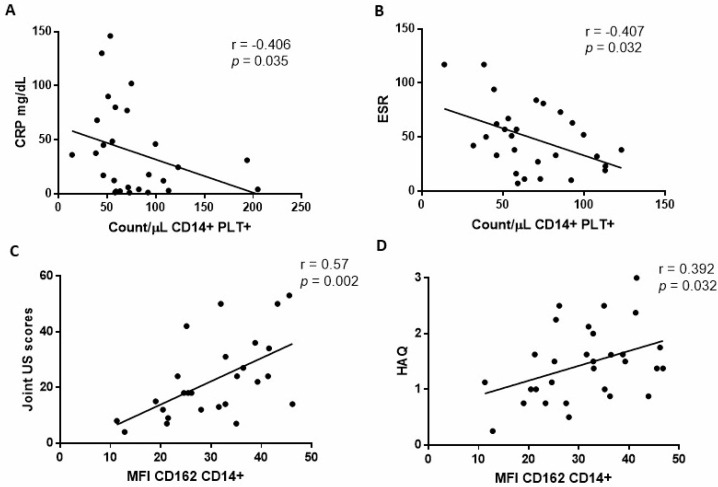
Association of monocytes with bound platelets and CD162 expression on monocytes with baseline clinical and laboratory parameters in rheumatoid arthritis (RA) patients. (**A**) Absolute number of monocytes with bound platelets (CD14+PLT+) correlation with CRP and (**B**) ESR. (**C**) CD162 expression on monocytes correlation with joint ultrasound (US) scores and (**D**) health assessment questionnaire (HAQ). Statistical analysis was performed using Spearman’s correlation.

**Figure 3 ijms-23-05748-f003:**
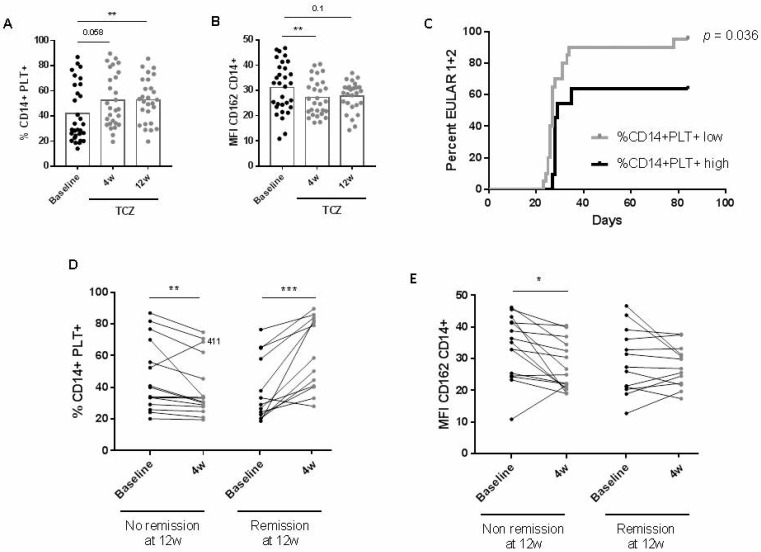
Changes in monocytes with bound platelets and CD162 expression after tocilizumab treatment and association with clinical response in rheumatoid arthritis (RA) patients. (**A**) Percentage of monocytes with bound platelets (CD14+PLT+) and (**B**) CD162 expression on monocytes before (Baseline) and four weeks (4 w) and twelve weeks (12 w) after tocilizumab (TCZ) treatment. (**C**) Kaplan-Meier curve for EULAR response. Percentage of patients with low or high %CD14+PLT+ achieving 1 (moderate) or 2 (good) EULAR response (days). (**D**) Changes in the percentage of CD14+PLT+ and (**E**) CD162 expression after 4 w of TCZ treatment in rheumatoid arthritis patients with no remission or remission at 12 w (**left**). Black circles represent RA patients before TCZ treatment, and grey circles RA patients after TCZ treatment. Statistical analysis was performed using Wilcoxon test for comparisons between baseline and 4 w or 12 w after TCZ treatment and log-rank Mantel–Cox test for the analysis of EULAR response during follow-up. * *p* < 0.05, ** *p* < 0.01, *** *p* < 0.001.

**Figure 4 ijms-23-05748-f004:**
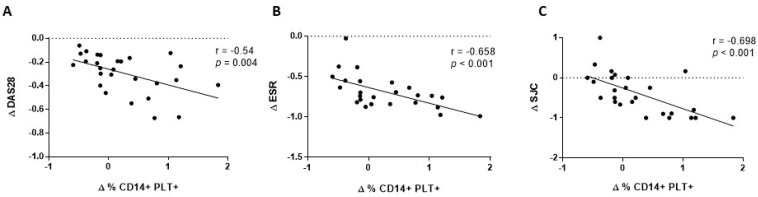
Association of changes in monocytes with bound platelets and CD162 expression on monocytes after 4 weeks of tocilizumab treatment with clinical and laboratory parameters. (**A**) Changes in percentage of monocytes with bound platelets (CD14+PLT+) correlation with changes in DAS28, (**B**) ESR and (**C**) swollen joint count (SJC). Statistical analysis was performed using the Spearman’s correlation.

**Figure 5 ijms-23-05748-f005:**
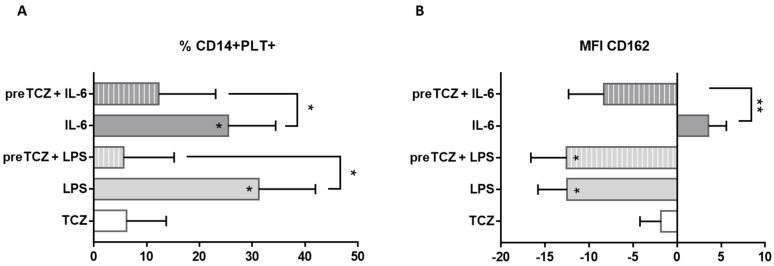
Effect of IL-6 and tocilizumab on platelet binding and CD162 expression on monocytes. Peripheral whole blood from healthy donors was cultured without stimulus or stimulated with lipopolysaccharide (LPS), recombinant human IL-6 or tocilizumab (TCZ). When TCZ stimulus is prior to LPS or IL-6 treatment is indicated as pre-. Differences between each condition and basal condition were calculated from 7 independent experiments. X axis represents percentage of increase or decrease from basal condition. Data are expressed as mean with standard error of mean. Stars inside the bars indicate comparison with basal condition. (**A**) Percentage of monocytes with bound platelets or (**B**) CD162 expression on monocytes was evaluated. Statistical analysis was performed using Wilcoxon test. * *p* < 0.05, ** *p* < 0.01.

**Table 1 ijms-23-05748-t001:** Demographic, clinical, and laboratory characteristic data of study patients.

	RA (*n* = 35)	HD (*n* = 15)	*p*
Sex, % (*n*), women	80.6 (28)	80 (12)	0.96
Age in years, mean ± sd	53.82 ± 10.72	52.48 ± 8.1	0.72
Years of evolution, median (IQR)	12.5 (7–17)		
Corticoids, % (*n*)	67.7 (21)		
Monotherapy, % (*n*)	29 (9)		
MTX, % (*n*)	42 (13)		
Previous biological therapies, % (*n*)	64.5 (20)		
HAQ, mean ± sd	1.4 ± 0.67		
DAS28, mean ± sd	5.7 ± 1.15		
ESR (mm/h), mean ± sd	51.4 ± 32		
CRP (mg/L), median (IQR)	24.6 (4–68)		
ACPA+, % (*n*)	61.3 (19)		
ACPA (UI/mL), median (IQR)	140.7 (54–318)		
RF+, % (*n*)	71 (22)		
RF (UI/mL), median (IQR)	161 (32.5–333.5)		

## Data Availability

Not applicable.

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
