# Peer review of "Increase of Circulating Monocyte–Platelet Conjugates in Rheumatoid Arthritis Responders to IL-6 Blockage"

_ijms, 2022, doi:10.3390/ijms23105748_

Round 1
Reviewer 1 Report
This manuscript reported that more monocytes bound to PLTs and had decreased mCD162 expression in RA patients. The percentage of CD14+PLT+ increased in RA patients who achieved remission after 12 weeks post-TCZ treatment. Patients with a lower percentage of CD14+PLT+ before TCZ treatment are more likely to achieve EULAR-defined response.
The data in this paper are interesting and might indicate the role of CD14+PLT+ as a potential biomarker in response prediction of TCZ. However, the lack of mechanism is a leading weakness for this study. The authors did not show any experimental explanation for the possible involvement of CD14+PLT+ in TCZ treatment. The involvement of mCD162 in this process is also unclear.
Author Response
We have already published mechanistic effects of platelets on monocytes in other chronic inflammatory diseases. Particularly, in systemic lupus erythematosus and ulcerative colitis, we have found that platelets increased the expression of costimulatory markers and IL-10 secretion (1,2). Despite we did not analyse these mechanisms in RA, it is tempting to speculate that comparable mechanisms are operating. The actual work is more focused on the potential usefulness of CD14+PLT+ as a marker of TCZ response. However, we agree the lack of mechanism is a leading weakness of the present study. We have included it as a limitation of this work in the last paragraph of the discussion.
- Mariscal, A.; Zamora, C.; Magallares, B.; Salman-Monte, T.C.; Ortiz, M.À.; Díaz-Torné, C.; Castellví, I.; Corominas, H.; Vidal, S. Phenotypic and Functional Consequences of PLT Binding to Monocytes and Its Association with Clinical Features in SLE. Int. J. Mol. Sci. 2021, 22, doi:10.3390/ijms22094719.
- Zamora, C.; Canto, E.; Nieto, J.C.; Garcia-Planella, E.; Gordillo, J.; Ortiz, M.A.; Suarez-Calvet, X.; Perea, L.; Julia, G.; Juarez, C.; et al. Inverse Association Between Circulating Monocyte-Platelet Complexes and Inflammation in Ulcerative Colitis Patients. Bowel Dis. 2018, 00, 1–11, doi:10.1093/ibd/izx106.
Reviewer 2 Report
This is an interesting and well-written manuscript with the goal of understanding whether a new mechanism of action of Tocalizumab on RA is through the regulation of the binding of platelets to monocytes. The experiments are well designed and presented and the conclusions are supported by the data. Only concern is the small number of patient samples and controls, especially since they have chosen to evaluate patients who have failed any DMARDs. Is this generalizable to early RA patients, or patients who have failed other treatments. Certainly the study as is sets the stage for evaluating these types of questions.
Author Response
We agree that it is small number of patients and controls. There are two main reasons. First, at the time of the study, tocilizumab was administered to patients who have previously failed DMARDs. In addition, in order to have a homogeneous cohort, patients were selected with long disease duration. To highlight this limitation, we have completed the statement in the last paragraph of discussion.
Reviewer 3 Report
The authors of the article ijms-1695041 entitled “Increase of circulating monocyte-platelet conjugates in rheumatoid arthritis responders to IL-6 blockage", investigated the levels of monocyte-PLT aggregates in patients with RA receiving IL-6 blockage treatment. Although this is a rather interesting article, there are a few questions that require attention.
Major comments:
1) Firstly, the authors should not use the term “tendency” to describe almost significant but non-significant statistical findings (Results section paragraph 2.2). No statistical significance means there are no differences even if the value is close to statistical significance. This should be changed throughout the manuscript. There a plenty of methodological papers that deal with the misuse of such terminology to describe non-significant findings.
2) The authors findings demonstrate that monocyte-PLT aggregates are increased in patients with RA compared to controls. Moreover, the aggregates increase further after usage of TCZ both in 4 and 12 weeks after treatment in patients with RA. This demonstrates that the initial high levels actually increase after IL-6 blockage. How can that be related to the disease pathophysiology? Is the explanation that it is just a “protective” mechanism enough. The authors need to discuss this more thoroughly.
3) Moreover, in their in vitro model the authors demonstrate that IL-6 addition in whole blood generates high number of monocyte-PLT aggregates (Figure 5). In the particular figure (Figure 5) the authors use their untreated blood as baseline for controls. Although this is a valid way to demonstrate their results, it would be recommended if the authors use actual MFI values in a bar chart. Furthermore, the authors’ in vitro findings are puzzling. Interestingly preincubation with TCZ attenuates the IL-6 dependent increase in monocyte-PLTs. However, this is the exact opposite compared to their ex vivo findings. The authors comment on this finding briefly in their discussion. It is suggested to comment further on this finding and try to explain it with relevant literature since these findings goes against their ex vivo findings and does not support their claim that this is relevant to the disease pathophysiology. Also it should be included as a limitation of the study.
Author Response
1) Accordingly with your comment we have removed “tendency” throughout the manuscript (3 comparisons).
2) We have included two sentences in the 7th paragraph of discussion describing this fact and proposing an explanation.
3) We have included the requested values for the in vitro model as supplementary data (Table S1).
4) We cannot consider the TCZ preventive effect observed in vitro a contradiction with in vivo observation, since RA patients already have higher levels of IL-6 when TCZ treatment is started. So we do not have an ex vivo situation equivalent to in vitro pre-incubation with TCZ to compare with. However, TCZ treatment after IL-6 does not change CD14+PLT+ levels in HD. We are not able to discriminate if this discordance with ex vivo observation of CD14+PLT+ increase after TCZ in RA patients is related to the origin of the cells (HD vs RA) dosage or time differences. As in vitro reproduction of the in vivo scenario is almost impossible, those puzzling findings are within what is expected, and it is not the first time it occurs in the literature (1, 2).
However, as we have included in the discussion (7th paragraph), in vitro experiments with RA samples will discriminate which differences observed in vitro and ex vivo are related to intrinsic properties of the cells and which are related to other factors.
- Wright, H. L., Cross, A. L., Edwards, S. W., & Moots, R. J. (2014). Effects of IL-6 and IL-6 blockade on neutrophil function in vitro and in vivo. Rheumatology (United Kingdom), 53(7), 1321–1331. https://doi.org/10.1093/rheumatology/keu035
- Chen, C. M., Li, S. C., Chen, C. Y. O., Au, H. K., Shih, C. K., Hsu, C. Y., & Liu, J. F. (2011). Constituents in purple sweet potato leaves inhibit in vitro angiogenesis with opposite effects ex vivo. Nutrition, 27(11–12), 1177–1182. https://doi.org/10.1016/j.nut.2011.01.005
Round 2
Reviewer 1 Report
The authors have addressed my concerns.
Reviewer 3 Report
The revised manuscript has been improved.